# Effects of Positive Fighting Experience and Its Subsequent Deprivation on the Expression Profile of Mouse Hippocampal Genes Associated with Neurogenesis

**DOI:** 10.3390/ijms24033040

**Published:** 2023-02-03

**Authors:** Olga E. Redina, Vladimir N. Babenko, Dmitry A. Smagin, Irina L. Kovalenko, Anna G. Galyamina, Vadim M. Efimov, Natalia N. Kudryavtseva

**Affiliations:** 1Federal Research Center, Institute of Cytology and Genetics, Siberian Branch of Russian Academy of Sciences, Novosibirsk 630090, Russia; 2Pavlov Institute of Physiology, Russian Academy of Sciences, Saint Petersburg 199034, Russia

**Keywords:** hippocampus, neurogenesis, daily agonistic interactions, chronically winning mice, fighting deprivation, RNA-Seq

## Abstract

The hippocampus is known as the brain region implicated in visuospatial processes and processes associated with learning and short- and long-term memory. An important functional characteristic of the hippocampus is lifelong neurogenesis. A decrease or increase in adult hippocampal neurogenesis is associated with a wide range of neurological diseases. We have previously shown that in adult male mice with a chronic positive fighting experience in daily agonistic interactions, there is an increase in the proliferation of progenitor neurons and the production of young neurons in the dentate gyrus (in hippocampus), and these neurogenesis parameters remain modified during 2 weeks of deprivation of further fights. The aim of the present work was to identify hippocampal genes associated with neurogenesis and involved in the formation of behavioral features in mice with the chronic experience of wins in aggressive confrontations, as well as during the subsequent 2-week deprivation of agonistic interactions. Hippocampal gene expression profiles were compared among three groups of adult male mice: chronically winning for 20 days in the agonistic interactions, chronically victorious for 20 days followed by the 2-week deprivation of fights, and intact (control) mice. Neurogenesis-associated genes were identified whose transcription levels changed during the social confrontations and in the subsequent period of deprivation of fights. In the experimental males, some of these genes are associated with behavioral traits, including abnormal aggression-related behavior, an abnormal anxiety-related response, and others. Two genes encoding transcription factors (*Nr1d1* and *Fmr1*) were likely to contribute the most to the between-group differences. It can be concluded that the chronic experience of wins in agonistic interactions alters hippocampal levels of transcription of multiple genes in adult male mice. The transcriptome changes get reversed only partially after the 2-week period of deprivation of fights. The identified differentially expressed genes associated with neurogenesis and involved in the control of a behavior/neurological phenotype can be used in further studies to identify targets for therapeutic correction of the neurological disturbances that develop in winners under the conditions of chronic social confrontations.

## 1. Introduction

Hippocampal neurogenesis has been shown to persist throughout life [1,2,3,4]; however, it is impaired in many psychiatric and neurological disorders [5,6], such as Alzheimer’s disease [7,8,9,10,11,12], schizophrenia [3,13], epilepsy [14], autism spectrum disorders [15], Parkinson’s disease [16,17], depression, anxiety [18,19,20,21,22], multiple sclerosis [23], motivated [24] and affective [25,26] behaviors, and dysfunction associated with stress [27,28]. Some of these diseases, studied both in humans and animals, are linked with a decrease in neurogenesis, whereas others are associated with its increase. The observed differences in the dynamics of adult neurogenesis may be related to various causes of the manifestation of the pathologies in question. The dynamics of the development of neurogenesis in the postnatal brain of mammals under normal and pathological conditions have been presented in many reviews in recent years [2,29,30].

Neurogenesis is a complex and highly regulated multistep process that is modulated by various endogenous and exogenous factors [31]. During the development of a pathology, changes in neurogenesis—from stem cells to the formation of new neurons—are accompanied by the modulation of many regulatory processes at transcriptional and post-transcriptional levels. Numerous genes involved in these processes change their magnitude of expression, thereby modifying many signaling pathways that play a key role in determining the fate of progenitor neurons and newly formed migrating neurons. Molecular genetic processes participating in the changes in neurogenesis during the development of various pathological conditions and aging are being actively investigated [32]. Nonetheless, specific features of these processes during the development of each disease associated with neurogenesis modulation are still not fully understood.

We have previously shown that a repeated experience of aggression accompanied by a positive fighting experience in daily agonistic interactions leads to the development of a behavioral pathology similar to psychosis with signs of an addiction-like state (reviewed in [33,34]). We have reported that the repeated positive fighting experience in males causes them to persistently demonstrate increased aggression and stereotypic behaviors, enhanced aggressive motivation, elevated anxiety, hyperactivity, enhanced impulsivity, and impaired social recognition [35]. We have demonstrated that the positive fighting experience in daily agonistic interactions leads to significant alterations of the expression of monoaminergic, glutamatergic, opioidergic, and other genes in the main reward-related brain regions, including the ventral tegmental area, dorsal and ventral striatum, prefrontal cortex, and midbrain raphe nuclei [33,34,36]. When winning males are deprived of the opportunity to participate in further fights, they manifest elevated aggressiveness [26,37,38,39]. An analysis of the dorsal-striatum transcriptome in chronically winning males and those after 2 weeks of subsequent fighting deprivation has revealed that the deprivation period partially restores the transcriptome; however, a distinct opioid-related withdrawal effect is observed in the fighting-deprived males [40].

We have also researched some parameters of neurogenesis and neuronal activity in mice with the positive fighting experience in chronic social conflicts. Our results have shown that during the period of agonistic interactions and enhanced aggression, there is an increase in the proliferation of progenitor neurons and the production of young neurons in the dentate gyrus (in hippocampus), and these parameters remain modified in winners deprived of further fights [26]. In contrast to the stably increased neurogenesis, neuronal activity significantly goes up on day 10 of social confrontations, diminishes on day 21 to the level of controls, and then significantly decreases during the 2-week deprivation period [26].

All of the above indicates that the model of repeated aggression and fighting deprivation is suitable for studying mechanisms of neurogenesis activation in the hippocampus. Therefore, the aim of this work was to identify hippocampal genes associated with neurogenesis and involved in the formation of behavioral features in mice with the chronic experience of wins in everyday agonistic interactions and during the subsequent 2-week deprivation of aggression.

## 2. Results

### 2.1. Differences between Groups of Control and Experimental Mice

RNA-Seq was performed to analyze the gene expression profile in the hippocampus of three groups of male mice: chronically aggressive for 20 days (A20), chronically aggressive for 20 days followed by 2 weeks of fighting deprivation (AD), and intact (control) mice (C). Figure 1 illustrates the differences in the transcription profile of the hippocampus between the experimental and control animals. As expected, the greatest differences (Euclidean distances) in the gene expression profile were observed between male winners and controls (comparison “C_A20”). The 2-week fighting deprivation partially reduced the differences in the hippocampal gene expression profile between winners and control animals (comparison “C_AD”), suggesting that many genes tend to restore their transcriptional levels during the 2-week deprivation period.

### 2.2. C_A20 Differentially Expressed Genes (DEGs) Associated with Neurogenesis

Everyday agonistic interactions for 20 days caused changes in transcription levels of 2258 genes in the hippocampus of winning males. According to the Mammalian Adult Neurogenesis Gene Ontology (MANGO) [32] database, in which 397 genes are annotated that are associated with the regulation of adult hippocampal neurogenesis, transcription levels of 72 DEGs associated with neurogenesis were altered in the hippocampus of winner mice (A20) (Appendix A). Most of these DEGs (70.8%) decreased their level of transcription under these experimental conditions. Among the 72 DEGs related to neurogenesis, 13 DEGs encode transcription factors, eight of which manifested a reduction in transcription during the aggression experience accompanied by wins.

#### Kyoto Encyclopedia of Genes and Genomes (KEGG) Analysis of C_A20 DEGs Associated with Neurogenesis

Functional annotation of the 72 neurogenesis-associated C_A20 DEGs was performed in the KEGG database. A complete list of the identified metabolic pathways and their related DEGs is presented in Appendix A, while Figure 2 shows the most enriched terms (false discovery rate [FDR] <5.00 × 10^−2^), among which “neuroactive ligand-receptor interaction” is the most significantly enriched (FDR = 2.95 × 10^−7^). Many DEGs are associated with pathways of neurodegeneration—multiple diseases, including Alzheimer’s disease. Figure 2 indicates that the transcriptome changes affected functions of dopaminergic, glutamatergic, and cholinergic synapses and various signaling pathways. Several DEGs related to long-term potentiation were also identified, highlighting the presence of changes in the strength of synaptic connections between hippocampal neurons, thus possibly indicating disturbances in learning and memory.

### 2.3. DEGs Associated with Neurogenesis and Making the Most Significant Contribution to Intergroup Differences (Comparison C_A20)

To identify neurogenesis-associated DEGs that make the greatest contribution to C_A20 differences, partial least squares discriminant analysis (PLS-DA) (Figure 3a) was used, followed by calculation of the correlation between gene expression and PLS-DA Axis 1 (Figure 3b). The genes characterized by the highest correlation coefficients can be regarded as the genes whose expression differences between control and A20 mice make the greatest contribution to the intergroup differences.

The analysis of the C_A20 DEGs associated with neurogenesis and making the largest contribution to between-group differences is presented in Appendix A (see column “correlation between gene expression and PLS-DA Axis 1”), and summarized data are shown in Table 1. In this table, readers can see that a large number of DEGs associated with neurogenesis seem to make a significant contribution to the intergroup differences. Many of these DEGs code for transcription factors.

To identify the C_A20 DEGs associated with neurogenesis and capable of contributing to the formation of abnormal emotion/affect behavior, we used the Rat Genome Database. The following terms were chosen, which, from our point of view, most appropriately characterize the behavior of winning mice: abnormal aggression-related behavior (including increased aggression), abnormal anxiety-related response, abnormal fear-related response, and abnormal response to social novelty. In addition, because functions of the hippocampus are primarily linked with learning and memory, we analyzed the terms learning or memory and abnormal learning/memory/conditioning. The association of DEGs with these terms is depicted in Appendix A, and the summarized data are presented in Table 1, which shows that the largest number of DEGs is associated with the term “abnormal anxiety-related response.” 

DEGs associated with behavior and making the largest contribution (r > |0.90|) to between-group differences are listed in Table 2. Four of the six DEGs in Table 2 encode transcription factors. One of them (*Nr1d1*, nuclear receptor subfamily 1, group D, member 1) features the highest coefficient of correlation between DEGs’ expression and PLS-DA Axis 1 and deals with an abnormal anxiety-related response. This finding indicated its significant role in the formation of intergroup differences in this trait.

According to the data presented in Table 2, special attention should also be paid to the *Fmr1* (fragile X messenger ribonucleoprotein 1) gene. Intergroup differences in the transcription level of the *Fmr1* gene were found to be a key factor for the manifestation of several behavioral traits: abnormal aggression-related behavior (including increased aggression), abnormal fear-related response, abnormal response to social novelty, and abnormal learning/memory/conditioning. Accordingly, *Fmr1* can be considered one of the main candidate genes related to the behavior of males with the chronic experience of wins in agonistic interactions. Characteristics of genes that may contribute the most (r >|0.90|) to the intergroup differences are shown in Table 3.

### 2.4. C_AD DEGs Associated with Neurogenesis

Comparison C_AD of hippocampal transcription profiles suggested that after the 2-week period of aggression deprivation, the expression of 795 genes significantly differed from that in the hippocampus of control mice. According to the information available in the MANGO database, among the C_AD DEGs, 31 genes are associated with neurogenesis (Appendix A). Most of these DEGs (80.6%) have weaker transcription as compared to the control. Six out of thirty-one neurogenesis-associated DEGs encode transcription factors, four of which have a reduced transcription level as compared to the control.

#### 2.4.1. KEGG Analysis of C_AD DEGs Associated with Neurogenesis

Although the period of deprivation caused a decrease in the number of DEGs, the results of DEG functional annotation in the C_AD comparison revealed that the functioning of many metabolic pathways in the hippocampus of AD mice remained altered as compared to controls (Figure 4). As in the group of 72 neurogenesis-associated C_A20 DEGs, metabolic pathway “neuroactive ligand-receptor interaction” remained the most significantly enriched (FDR = 9.79 × 10^−4^) after the deprivation period. C_AD DEGs associated with neurogenesis and related to the identified metabolic pathways are listed in Appendix A.

STRING analysis showed that in the hippocampus of mice after the 2-week deprivation of social conflicts, the proteins encoded by most of the DEGs associated with neurogenesis take part in signaling (Figure 5), and many of them are associated with a response to stress. These data suggested that after the 2-week deprivation period, the physiological state of the hippocampus of experimental mice still “remembered” the influence of the social confrontations, one of whose characteristics is stress.

#### 2.4.2. Changes in C_AD DEGs’ Transcription during the Fighting Deprivation as Compared with the Period of Agonistic Interactions

Many C_AD DEGs (24 out of 31) associated with neurogenesis significantly changed their level of transcription during the 20-day confrontations and did not restore it during the deprivation period (Figure 6). Consequently, it can be theorized that the differential transcription of these genes may be related to processes that arise during the period of social confrontations and remain unchanged during the period of fighting deprivation. 

The expression of the remaining seven DEGs changed in different ways during the experiment (Figure 7). The expression of the *Egr1* gene significantly declined only during the fighting deprivation, and this alteration may be explained by the cessation of social confrontations, i.e., by changes in experimental conditions. 

The expression of genes *Htr2c* and *Prlr* during the period of deprivation partially normalized but remained significantly lower as compared to the control. Accordingly, it is likely that the expression of these genes is associated with the factors whose influence significantly weakened but remained substantial during the fighting deprivation.

The expression of the remaining four genes (*Epha5*, *Notch3*, *Slc7a11*, and *Mif*) gradually changed throughout the experiment and became significantly different from the control level only after the period of fighting deprivation. These genes may be related to processes that determine the experimental mice’s phenotypic traits that appeared during the period of intermale confrontations and were exacerbated during the period of the fighting deprivation. 

#### 2.4.3. DEGs Associated with Neurogenesis and Making the Most Significant Contribution to C_AD Differences 

PLS-DA (Figure 8a) followed by analysis of the correlation between gene expression and PLS-DA Axis 1 (Figure 8b) was used to identify the DEGs that make the greatest contribution to C_AD differences. The genes having the highest correlation coefficients can be regarded as the genes whose expression differences between control and experimental mice make the greatest contribution to the intergroup differences. The result of the analysis is presented in Appendix A (column “correlation between gene expression and PLS-DA Axis 1”), and summarized data are shown in Table 4. According to latter table, it is likely that most DEGs associated with neurogenesis make a significant contribution to C_AD differences. Table 5 lists characteristics of the three neurogenesis-associated DEGs (*Nr1d1, Fmr1,* and *Creb1*) that encode the transcription factors that contribute the most (r > |0.90|) to the intergroup differences. 

The neurogenesis- and behavior-associated DEGs that may contribute the most (r > 0.90) to C_AD differences are represented by three genes: *Nr1d1*, *Fmr1*, and *Pten* (Table 6). Thus, even after the 2 weeks of fighting deprivation, DEGs *Nr1d1* and *Fmr1* encoding transcription factors could be considered key genes that played a significant part in the formation of intergroup differences in the behavioral traits under study.

### 2.5. Identification of New Candidate Genes Related to Behavior 

Table 6 presents genes that are associated with neurogenesis and can be regarded as playing a significant role in the formation of C_AD differences and as associated with behavioral traits that characterize the phenotype of the experimental mice. We believe that the DEGs coexpressed with them may also participate in the manifestation of these traits, even though these genes have not yet been implicated in the behavioral traits under study. Analysis of correlation between the expression of genes *Nr1d1*, *Fmr1*, and *Pten* and the expression of all 795 C_AD DEGs showed that the expression of *Nr1d1* correlates (r > 0.823; *p* < 0.001) with the expression of 314 DEGs, *Fmr1* correlates with 358 DEGs, and *Pten* with 383 DEGs (Appendix A), indicating highly correlated changes in transcription levels of many DEGs found in the hippocampus of AD mice. The top 10 coexpressed DEGs characterized by the highest correlation coefficients are presented in Table 7. In Appendix A, which lists the characteristics of their expression, one can see that the alterations of the expression of most of these DEGs can make a significant contribution to C_AD differences. 

Among the genes coexpressed with *Fmr1*, genes *Spred1* and *Pten* have already been characterized as genes associated with an abnormal anxiety-related response (Rat Genome Database). In addition, *Pten* is related to abnormal aggression-related behavior and to an abnormal fear-related response. From the obtained correlations, we can conclude that the genes presented in Table 7 may be involved in an abnormal anxiety-related response, and the genes whose expression correlates only with *Fmr1* expression may also be associated with one or more of the behavioral traits under consideration: abnormal aggression-related behavior, abnormal fear-related response, abnormal response to social novelty, and abnormal learning/memory/conditioning.

## 3. Discussion

Previously, it has been reported that in the hippocampus of male mice with a chronic experience of wins in agonistic interactions, neurogenesis strengthens by day 10 and remains at the same high level until day 20 of social confrontations and during the subsequent 2-week deprivation of fighting [26]. To identify DEGs that may be associated with changes in hippocampal neurogenesis, we conducted a comparative study on hippocampal transcriptomes of adult male mice with a chronic experience of wins before and after a 2-week deprivation of conflicts and a control group of males. The experience of aggression accompanied by wins within 20 days caused significant changes in levels of transcription of many genes, 72 of which, according to specialized database MANGO [32], are related to neurogenesis in the hippocampus. Among them, 13 DEGs encode transcription factors, four of which (*Nr1d1*, *Fmr1*, *Atf2*, and *Fxr2*) were found to be the DEGs that were likely to contribute the most to between-group differences “C_A20” (Table 2). These transcription factor genes are associated with several behavioral traits that specifically describe the phenotypic characteristics of the 20-day winners. 

The largest number of DEGs associated with neurogenesis and making the largest contribution to C_A20 differences are associated with an abnormal anxiety-related response (*Nr1d1*, *Fmr1*, *Atf2*, and *Pten*). Earlier studies using the same experimental design have shown that repeated positive fighting experience enhances aggressive behavior in the winning males and heightens their anxiety, as evidenced by an increase in several parameters of anxiety-driven behavior in the plus-maze test [26,41]. Accordingly, we can hypothesize that the changes in the expression of all genes presented in Table 2 are a significant factor for the formation of the anxious phenotype in the male winners. 

After the 2-week fighting deprivation period, the number of DEGs declined significantly because the transcription of many genes returned to the control level. By contrast, the magnitude of hippocampal transcription of *Nr1d1, Fmr1*, and *Pten* did not change during the fighting deprivation and remained significantly different between AD and control mice. These genes, which presumably make the largest contribution to the C_AD differences (Table 6), can be considered key genes involved in the formation of behavioral features of AD males. 

Because neurogenesis increases in the hippocampus of winning males during the 20 days of agonistic interactions and remains strong during the 2-week period of deprivation [26], it is logical to expect similar dynamics of transcription of genes that may participate in this process. As follows from Figure 6, this is how one can describe changes in transcription levels of most (24 out of 31) DEGs associated with neurogenesis in the C_AD comparison. These include *Nr1d1*, *Fmr1*, and *Pten*, allowing us to hypothesize that these genes may play a key role in the regulation of processes related to the enhancement of neurogenesis under our experimental conditions of intermale confrontations. 

The *Nr1d1* gene encodes nuclear receptor subfamily 1, group D, member 1. In most research articles, the function of this gene is assigned to the molecular circadian clock system. Dysregulation of circadian rhythms is often considered a key symptom of mood or anxiety disorders [42]. According to clinical observations, patients with mood disorders often have the abnormalities in a circadian rhythm that correlate with nucleotide substitutions in various circadian clock genes including *NR1D1* [43]. It has been shown that *Nr1d1* is one of the circadian genes whose expression may be modulated by a therapeutic dose of lithium [44], and this finding may be relevant to the treatment of anxiety in neurological disorders [45]. Our previous results have revealed that lithium can also modulate anxiety in male mice with repeated experience of aggression [46]. On the basis of those observations, we can theorize that the *Nr1d1* gene is involved in the control of the anxiety level in males with a positive chronic fighting experience. 

The association of *Nr1d1* with adult hippocampal neurogenesis in mice was suggested when an experimental deletion of the *Rev-erb-alpha* (*Nr1d1*) gene accelerated the proliferation of hippocampal neurons [47]. The loss of *Rev-erb-alpha* also caused alterations in memory and mood-related behaviors [47], thereby confirming the involvement of *Nr1d1* in the formation of abnormal emotion/affect behavior. According to our earlier experiment, neurogenesis in the hippocampus of aggressive males also increases [26], but *Nr1d1* expression also significantly goes up during the period of daily social conflicts and remains increased during the subsequent period of deprivation of aggression. NR1D1 is known as a constitutive transcriptional repressor. Accordingly, when it is lost in knockout animals, it can be assumed that all genes under its control are activated. Under the conditions of our current experiment, significant changes in transcription levels of ribosomal genes in different parts of the brain have been described, and activation of ribosomal genes is reported in the hippocampus of males with a positive fighting experience [48]. This finding indicates significant changes in hippocampal translation processes, which may also be related to the induction of neurogenesis. Perhaps in our experiment, the increase in *Nr1d1* gene transcription serves to curb excess neurogenesis, which is stimulated by the experimental conditions. It is also possible that the upregulation of *Nr1d1* transcription is associated with changes related to the emotional state of male winners. This notion is in good agreement with the results of a work showing that the expression of several circadian-clock genes including *Nr1d1* is high in the hippocampus of subjects with substance use disorder [49]. Our previous studies indicate that a repeated positive fighting experience in male mice results in the emergence of signs of addiction-like behavior [38] and in changes of the expression of genes associated with drug addiction in reward-related brain regions [34]. 

The relation of *Nr1d1* activation with the formation of addiction-like behavior and its key role in these processes is also observed in studies on other areas of the brain. It has been reported that *Nr1d1* that is induced in the mouse striatum by chronic drug administration has the most substantial impact on the gene expression profile [50]. In our recent paper, *Nr1d1* also proved to be a prime candidate for participation in the gene networks related to transcriptional repression in the dorsal striatum of AD mice compared to controls [40]. Of course, the features of *Nr1d1* expression can be associated with various processes in the brain, but its activation in the brain areas that are usually not associated with neurogenesis processes does not negate the involvement of *Nr1d1* in them, because results of research on other brain areas indicate that adult neurogenesis may also be stimulated by neurological processes outside the classic neurogenic niches [51]. 

Another gene that deserves special attention in our study is *Fmr1*, which codes for a translational regulator (fragile X messenger ribonucleoprotein 1). FMRP regulates gene expression and the translation of multiple mRNAs playing an important part in the development and maintenance of neuronal synaptic connections [52]. According to the Rat Genome Database, *Fmr1* is associated with several behavioral traits (abnormal aggression-related behavior, increased aggression, abnormal anxiety-related response, abnormal fear-related response, abnormal response to social novelty, and abnormal learning/memory/conditioning) that are relevant to our current work. The transcription of *Fmr1* proved to be downregulated in the hippocampus of both A20 and AD mice (Figure 6). The lack of fragile X mental retardation 1 protein (FMRP) causes fragile X syndrome (FXS) characterized by intellectual disability, autism, hyperactivity, and some other phenotypic anomalies [53]. *Fmr1* knockout mice exhibit a decreased corticosterone response to an acute stressor and reduced anxiety in open-field and elevated plus maze tests [54]. The loss of *Fmrp* promotes the proliferation of adult neural progenitor/stem cells, but it leads to reduced survival of young neurons in the dentate gyrus, thereby resulting in diminished neuronal differentiation but enhanced glial differentiation [55]. These processes have been linked to the ventral dentate gyrus: a hippocampal subregion more closely associated with emotion than the dorsal dentate gyrus is [54]. Therefore, we can hypothesize that downregulation of *Fmr1* transcription in the hippocampus of males with chronic experience of wins in social confrontations contributes to the observed increase in neurogenesis. The weakened transcription of the *Fmr1* gene persists during the 14-day fighting deprivation, in good agreement with experimental results about the level of neurogenesis in our earlier paper [26]. Nevertheless, as we saw above, the *Fmr1* underexpression associated with increased neurogenesis is also linked with reduced survival of newly generated neurons. Accordingly, the question of the survival of neurons in the hippocampus of experimental mice in the current study remains open. However, we can speculate that it may decline because our previous study has shown that the number of c-Fos–positive cells in the dentate gyrus (in hippocampus) significantly goes down during the period of fighting deprivation [26]. According to the results from ref. [54], a complete loss of *Fmr1* gene expression is accompanied by a decrease in anxiety. On the contrary, under the conditions of our experiment, the elevated level of anxiety in the winners did not decrease to the control level during the deprivation of fights [56]. The simplest explanation for these discrepancies, of course, is that in our experiment, many genes have an effect on the phenotype. 

Above we discussed two genes encoding transcription factors that, according to our analysis, may play a key role in the processes occurring in the hippocampus of mice with a chronic positive fighting experience and in the subsequent period of deprivation of confrontations. Their apparent opposite effects on neurogenesis point to their participation in the system’s striving to achieve physiological homeostasis. Nonetheless, earlier behavior-testing experiments and observations in this study indicate that during the fighting deprivation, only partial normalization occurs, and numerous genes, including those associated with neurogenesis, remain differentially expressed. These genes deal with signaling and stress responses (Figure 5), suggesting that the winning males experience stress that persists during the 2 weeks of fighting deprivation. 

In this work, we examined genes whose participation in neurogenesis in the hippocampus has been experimentally proven. On the other hand, the expression of these key genes correlated with the expression of many other hippocampal DEGs. In accordance with the notion that correlations may be indicative of some shared or indirect causal relations underlying trait manifestation [57], we propose that among these DEGs, there may also be genes taking part in the formation of behavioral features of males with chronic experience of wins in daily agonistic interactions.

Most DEGs associated with neurogenesis retained an altered transcription level during the period of fighting deprivation. Nonetheless, Figure 6 shows that some of them tend to return to the control level of transcription. Therefore, it is possible that some of them require a longer period to restore the baseline level of transcription. On the other hand, we can also theorize that some of these DEGs never return to baseline levels.

The pattern of changes in transcription levels of DEG clusters in the hippocampus observed in our study corresponds to a state of increased aggression, elevated anxiety, and some deviations of behavior related to learning and memory in experimental mice [33,35]. Having identified the most likely genes that make the greatest contribution to intergroup differences, we tried to assess their impact on the phenotype, taking into account the direction of change in their transcription in the hippocampus. Nevertheless, we understand that the phenotype of experimental mice, of course, arises as a set of changes in transcription profiles in different brain structures. At the same time, it should be kept in mind that in different regions of the brain, transcription of the same DEG can change in different directions, as described in our previously published articles [34,36].

Another key point is that our paper presents alterations of the gene expression profile in the hippocampus of mice after a long period of social confrontation. On the other hand, acute and chronic stressors are known to have different effects on gene expression in the brain [58]. Accordingly, it is possible that the transcriptional profile of genes in the hippocampus of the experimental mice during the first days of fighting differs from that after the 20-day period of daily agonistic interactions. Of course, a limitation of our study is that the determination of gene expression profiles was carried out only at two experimental time points: after 20 days of daily victories in social conflicts and then after 14 days of deprivation of fights.

The reader should also keep in mind that this work was performed at the transcriptional level, and mRNA expression may not necessarily reflect the corresponding protein levels. Nonetheless, as shown in many studies, transcriptomic analysis allows investigators to determine the main molecular events in cells and to understand which specific genes and metabolic pathways are most significantly involved in the adaptation of the body to effects of endogenous or exogenous factors. In this regard, the present authors hope that these results will be useful for further advancements of ideas about the molecular genetic basis of adult neurogenesis in the hippocampus under the conditions of psychosocial stress.

## 4. Materials and Methods

### 4.1. Animals 

The experiments were carried out using 10–12-week-old C57BL/6J male mice. The mice were kept in the Conventional Vivarium (federal research center Institute of Cytology and Genetics, SB RAS, Novosibirsk, Russia) under standard conditions at 22 ± 2 °C on a 12/12 h light–dark cycle (lights on at 8:00 AM) with dry laboratory feed and water available ad libitum. All procedures were conducted in compliance with European Communities Council Directive 210/63/EU of 22 September 2010. The study protocol was approved by the Bioethical Council of the federal research center Institute of Cytology and Genetics SB RAS (Novosibirsk, Russia), Protocol No. 613 dated 24 March 2010.

### 4.2. Creation of the Repeated Positive Fighting Experience for Male Mice by Agonistic Interactions 

The model used to induce chronic intermale confrontations has been described in detail in several papers [35,59]. The protocol of the current experiment is described in our previous article [40]. Figure 9 outlines the timetable of the experiment. 

Each pair of male mice was placed in a cage (28 × 14 × 10 cm) divided in half by a perforated transparent partition, which allowed the animals to hear, see, and smell each other but precluded physical contact. The animals were left alone for 2 or 3 days to adapt to unfamiliar housing conditions before the start of the agonistic interactions. The experiment was carried out at the same time (14:00–17:00 h local time) every day. The cage cover was first replaced with a transparent one, and after 5 min (the period required for the mice to start responding to a partner in the adjacent compartment), the partition was removed for 10 min to induce intermale social confrontations. The mouse that attacked, bit, and chased the opponent was considered the winner. The superiority of one of the males was established during two or three encounters with the same opponent. The male that displayed only defensive behaviors (upright postures, withdrawing, freezing, or lying on its back) was defined as the loser. The design of the experiment is aimed at ensuring that painful consequences of agonistic interactions (injuries) are absent in the defeated mice. In cases of strong (very active and prolonged), aggressive attacks, the partition was immediately restored to prevent injury to the defeated male. The defeated male (the loser) confronted the same winner for 3 days, and then every day after the trial, the defeated male was placed in an unfamiliar cage with an unfamiliar winner behind a partition. Each winner remained in his original cage. The intermale confrontation procedure was carried out once a day for 20 days. 

In each experiment, we monitored the behavior of all males by videotaping the behavior during the agonistic interactions. Observer XT software (version 7.0; Noldus Information Technology, Wageningen, The Netherlands) was employed for manual registration of the behavioral indicators during the test. This approach made it possible to identify the most aggressive mice, demonstrating the greatest number and duration of attacks, hyperactivity, and other relevant characteristics. Individuals with the most pronounced aggressive phenotype were selected for transcriptomic analysis. 

Three groups of mice were formed for the experiment: (1) Controls, i.e., mice without the experience of agonistic interactions; (2) winners, i.e., a group of aggressive mice chronically winning during 20 days in the daily agonistic interactions (A20); and (3) aggression-deprived mice (AD), i.e., the winners (A20) after a period of deprivation of fighting for 14 days. These three groups of animals, six mice per group, were subjected to the transcriptomic analysis.

The winners at 24 h after the last agonistic interaction, the control animals, and AD mice were decapitated simultaneously. The hippocampus was dissected by the same experimenter according to the map in the Allen Mouse Brain Atlas [60]. All hippocampal subregions from both hemispheres were collected. The samples were placed in an RNAlater solution (cat. #: AM7020; Life Technologies, Carlsbad, California, USA) and stored at −70 °C until sequencing.

### 4.3. RNA-Seq Data Collection and Processing

The frozen hippocampus samples (A20, AD, and control, *n* = 6 in each group) were delivered to JSC Genoanalytica (Moscow, Russia) for RNA-Seq sequencing. Total RNA was extracted using the PureLink RNA Micro Kit (cat. #: 12183016; Invitrogen, Waltham, MA, USA) according to the protocol of the manufacturer. A 2100 Bioanalyzer system (Agilent Technologies, Santa Clara, CA, USA) with the RNA 6000 Nano Kit (cat. #: 5067-1511; Agilent, USA) was used to estimate the quantity of RNA and the RNA integrity number (RIN). RINs of all samples were greater than 8.5. Next, 1 μg of total RNA was processed by means of the Dynabeads mRNA Purification Kit (cat. #: 61006; Ambion, Thermo Fisher Scientific, Waltham, MA, USA). For each tissue sample, all the extracted mRNA was used to create cDNA libraries with the help of the NEBNext mRNA Library PrepReagent Set for Illumina (cat. #: E7770S; New England Biolabs, Ipswich, MA, USA) according to the manufacturer’s protocol. Single-end sequencing of the cDNA libraries was performed on the Illumina HiSeq 2500 platform (Illumina Sequencing, San Diego, CA, USA) with a read length of 50 bases. The target coverage was set to 20 million reads per sample. All hippocampus samples were processed for each of the six animals in a group separately and analyzed as biological replicates. The raw reads from the RNA-seq experiments were trimmed for quality (phred ≥ 20) and length (≥32 bp) in Trimmomatic v.3.2.2 [61]. The sequencing data were mapped to the mouse reference genome (GRCm38.p3) available in GenBank using the STAR aligner [62]. Quality metrics of the mapped data (Appendix A) were determined in the Spliced Transcripts Alignment to a Reference (STAR) software [63]. Cuffnorm software [64] was employed for expression rate assessment in FPKM units.

### 4.4. Functional Annotation of the DEGs

The KEGG Pathway Database [65] was used to identify metabolic pathways significantly (*p* < 0.05) enriched within DEG sets. To determine the association of DEGs with a behavior/neurological phenotype, the Neurological Disease Portal (Phenotypes, Mouse) in the Rat Genome Database [66] was employed. The STRING database was used to construct the functional protein association network [67]. An atlas of combinatorial transcriptional regulation in mice and humans [68] was utilized to identify DEGs encoding transcription factor genes.

### 4.5. Statistical Methods 

The acquired RNA-Seq data (in FPKM values) were log_2_-transformed, centered, normalized, and scaled by principal coordinate analysis based on Euclidean metric distances. Then, PLS-DA was performed using the pattern of covariation for linear combinations between two blocks of variables [69]. These procedures resulted in the construction of PLS-DA Axes maximizing distances between the experimental and control mice. Next, calculation of Pearson correlation coefficients helped to detect a set of variables (expressed genes) that were expected to maximize the covariance between a fixed dummy matrix representing group membership (for experimental and control mice) and gene expression in these animals. The calculated correlation revealed the genes characterized by the most deviation along the first functionally meaningful synthetic axis (PLS-DA Axis 1). These genes were assumed to be candidates making the largest contribution to intergroup differences. The normalized RNA-Seq data were also used for Pearson correlation analysis. A 99.9% [df = 10; *p* < 0.001 (r = 0.823)] two-tailed confidence interval was considered significant. Software packages STATISTICA 12.0 (StatSoft, Tulsa, OK, USA) and JACOBI4 [70] were employed for the data analysis and presentation.

## Figures and Tables

**Figure 1 ijms-24-03040-f001:**
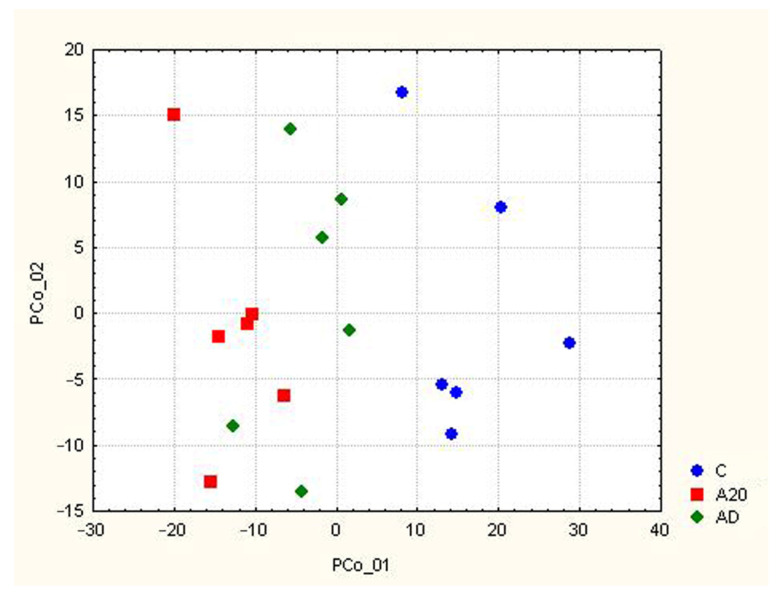
Differences in the transcription profile of the hippocampus between winners and control animals (principal coordinate analysis using Euclidean distances). C: Control mice without the experience of agonistic interactions; A20: males with consecutive 20 days of wins in the daily agonistic interactions; AD: A20 mice after subsequent 14 days of fighting deprivation.

**Figure 2 ijms-24-03040-f002:**
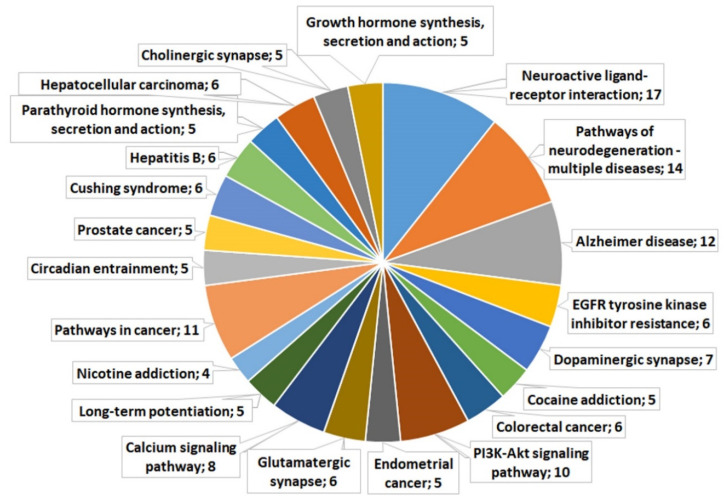
KEGG pathways for the 72 genes associated with neurogenesis and found to be differentially expressed in our comparison of the hippocampal transcription profile between control male mice and males with consecutive 20 days of wins in daily agonistic interactions.

**Figure 3 ijms-24-03040-f003:**
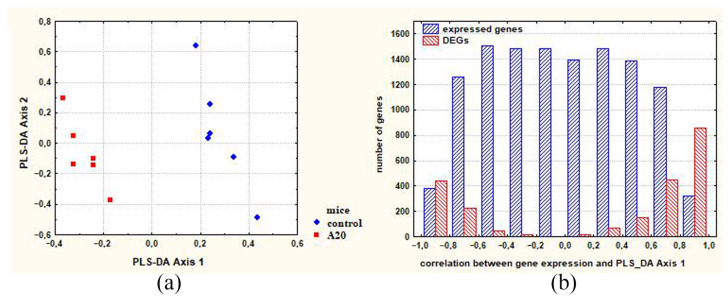
(**a**) Axes maximizing distances between control and A20 mice (males with consecutive 20 days of wins in daily agonistic interactions); (**b**) distribution of expressed genes along the axis representing the correlation between gene expression and PLS-DA Axis 1. DEGs: differentially expressed genes.

**Figure 4 ijms-24-03040-f004:**
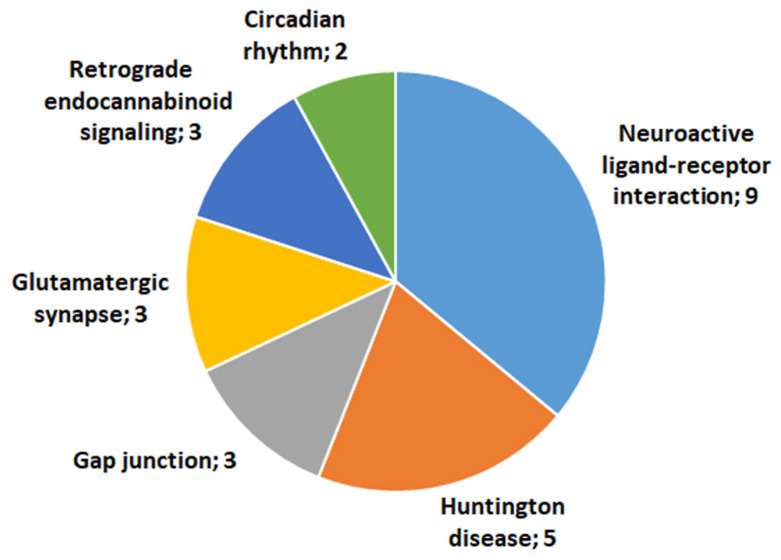
KEGG pathways related to the 31 genes associated with neurogenesis and found to be differentially expressed in our comparison of the hippocampal transcription profile between control male mice and males with consecutive 20 days of wins in daily agonistic interactions and then deprived of fighting for 14 days.

**Figure 5 ijms-24-03040-f005:**
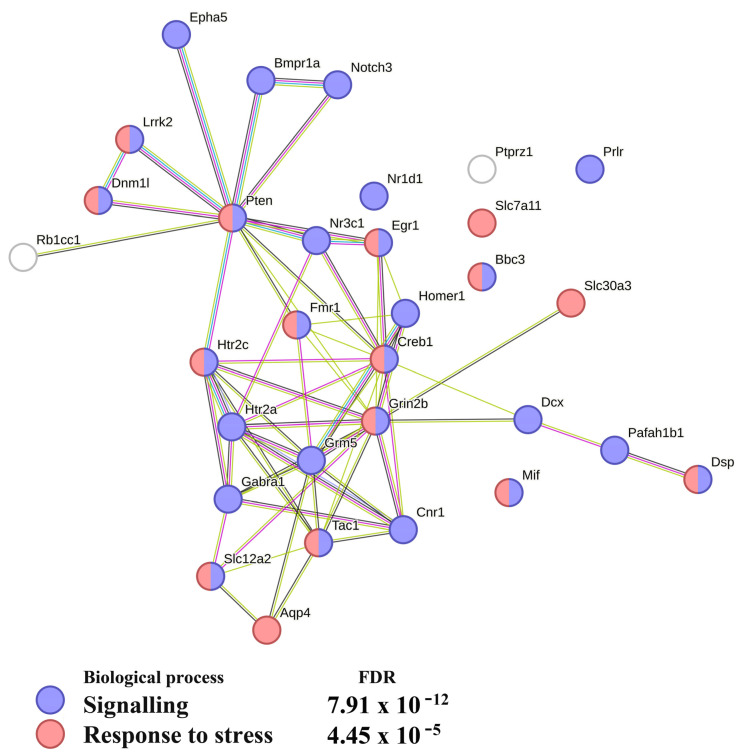
Functional enrichment analysis of the neurogenesis network constructed for the C_AD comparison. This analysis suggests that the DEGs in question may contribute to the signaling and response to stress. The functional enrichment network was constructed by means of the STRING database (https://string-db.org/; accessed on 7 November 2022) using the DEGs associated with neurogenesis. Each node represents all the proteins produced by a single protein-coding gene. Edges are protein–protein associations. Purple lines indicate experimentally determined interactions; blue lines denote known interactions from curated databases; dark blue lines represent gene co-occurrence; black lines indicate coexpression; and green lines represent results of text mining. Protein–protein interaction (PPI) enrichment *p*-value < 1.0 × 10^−16^. FDR: false discovery rate.

**Figure 6 ijms-24-03040-f006:**
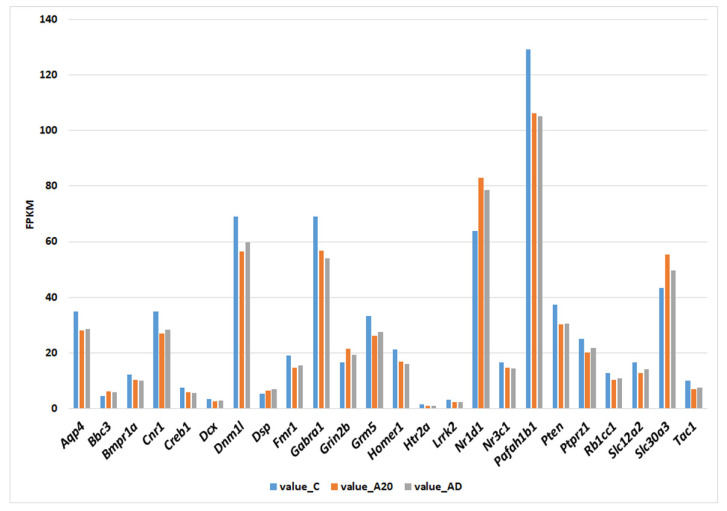
Genes that significantly changed their levels of transcription during the period of agonistic interactions (C_A20 DEGs), whose expression did not get restored during the fighting deprivation (C_AD differences are statistically significant). FPKM, fragments per kilobase of transcript per million mapped reads; value C: expression in control mice (no experience of agonistic interactions); value_A20: expression in males with consecutive 20 days of wins in daily agonistic interactions; value_AD: expression in males with consecutive 20 days of wins in daily agonistic interactions deprived of fighting for 14 days.

**Figure 7 ijms-24-03040-f007:**
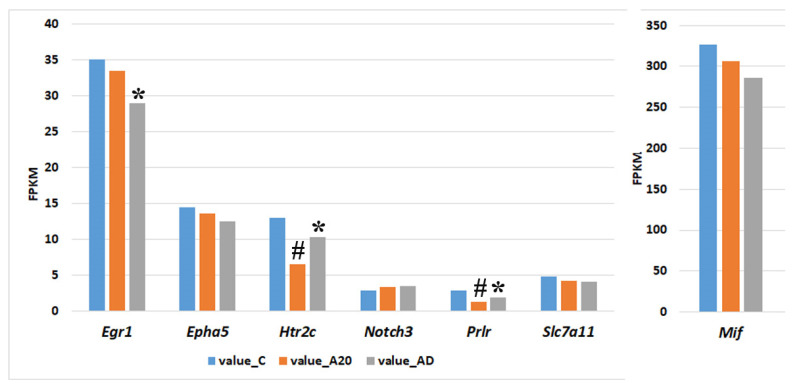
Neurogenesis-associated C_AD DEGs whose expression changed during the fighting deprivation. * DEGs in the A20_AD comparison; ^#^ DEGs in the C_A20 comparison. FPKM, fragments per kilobase of transcript per million mapped reads; value C: expression in control mice (no experience of agonistic interactions); value_A20: expression in males with consecutive 20 days of wins in daily agonistic interactions; value_AD: expression in males with consecutive 20 days of wins in daily agonistic interactions and then deprived of fighting for 14 days.

**Figure 8 ijms-24-03040-f008:**
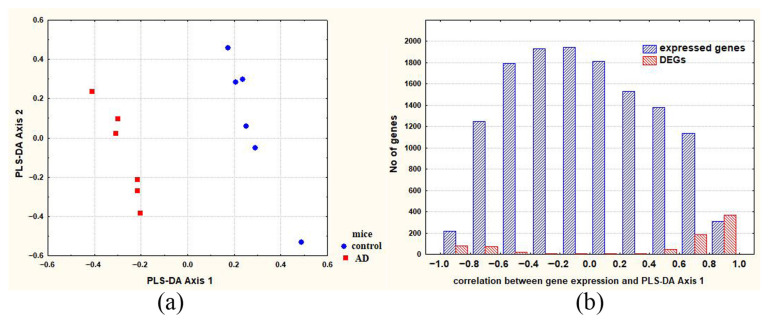
(**a**) Axes maximizing the distances between control and AD mice (males with consecutive 20 days of wins in daily agonistic interactions and then deprived of fighting for 14 days). (**b**) Distribution of expressed genes along the axis representing the correlation between gene expression and PLS-DA Axis 1. DEGs: differentially expressed genes.

**Figure 9 ijms-24-03040-f009:**
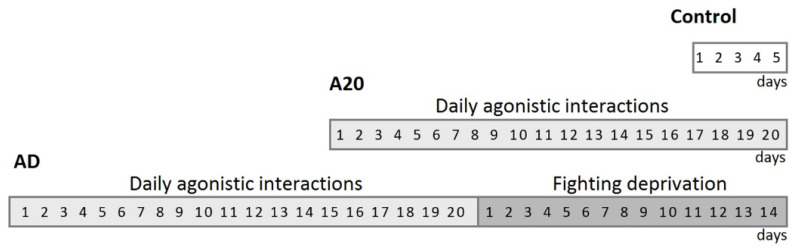
Graphical representation of the experiment. A20: males with consecutive 20 days of wins in daily agonistic interactions; AD: the A20 mice after subsequent 14 days of fighting deprivation; Control: mice without the experience of agonistic interactions. Control males were housed one per cage for 5 days, which enabled them to feel dominant and potentially able to demonstrate aggressive behavior in a conflict situation.

**Table 1 ijms-24-03040-t001:** Characterization of C_A20 DEGs associated with neurogenesis.

Correlation between DEGs’ Expression and PLS-DA Axis 1 |r|	No. of DEGs	Transcription Factor Genes	Abnormal Aggression- Related Behavior	Increased Aggression	Abnormal Anxiety-Related Response	Abnormal Fear-Related Response	Abnormal Response to Social Novelty	Learning or Memory	Abnormal Learning/ Memory/ Conditioning
1.00–0.90	15	4	3	2	4	3	1		1
0.89–0.80	19	2	2	2	5	2		1	
0.79–0.70	18	6			3	2			
0.69–0.60	8		1	1	3				
0.59–0.50	6	1			2	1		1	1
0.49–0.40	4				1			1	1
0.39–0.30									
0.29–0.20	1								
0.19–0.10									
0.09–0.00	1								

**Table 2 ijms-24-03040-t002:** Candidate genes associated with neurogenesis and behavioral traits and making the largest contribution to intergroup differences (comparison C_A20).

Correlation between DEGs’ Expression and PLS-DA Axis 1 |r|	Abnormal Aggression-Related Behavior	Increased Aggression	Abnormal Anxiety- Related Response	Abnormal Fear-Related Response	Abnormal Response to Social Novelty	Learning or Memory	Abnormal Learning/ Memory/ Conditioning
1.00–0.95			*Nr1d1 **				
0.94–0.90	*Fmr1 **	*Fmr1 **	*Fmr1 **	*Fmr1 **	*Fmr1 **		*Fmr1 **
	*Braf*	*Braf*	*Atf2 **	*Pten*			
	*Pten*		*Pten*	*Fxr2 **			

* Transcription factor genes.

**Table 3 ijms-24-03040-t003:** DEGs associated with neurogenesis and making the largest contribution (r > |0.90|) to intergroup differences in behavior.

Transcription Factor Genes	Gene Expression, FPKM		log_2_ (Fold Change) A20/C	q_Value	Correlation between Gene Expression and PLS-DA Axis 1	Gene Name
	Control	A20				
*Nr1d1 **	63.81	82.94	0.38	0.001	−0.957	Nuclear receptor subfamily 1, group D, member 1
*Fmr1 **	19.02	14.66	−0.38	0.001	0.944	Fragile X messenger ribonucleoprotein 1
*Atf2 **	31.49	26.75	−0.24	0.001	0.931	Activating transcription factor 2
*Braf*	21.25	18.12	−0.23	0.0026	0.919	Braf transforming gene
*Pten*	37.27	30.34	−0.30	0.0010	0.917	Phosphatase and tensin homolog
*Fxr2 **	46.90	53.13	0.18	0.019	−0.909	Fragile X mental retardation, autosomal homolog 2

* Transcription factor genes.

**Table 4 ijms-24-03040-t004:** Characterization of C_AD DEGs associated with neurogenesis.

Correlation between DEGs’ Expression and PLS-DA Axis 1 |r|	No. of DEGs	Transcription Factor Genes	Abnormal Aggression-Related Behavior	Increased Aggression	Abnormal Anxiety- Related Response	Abnormal Fear-Related Response	Abnormal Response to Social Novelty	Learning or Memory	Abnormal Learning/ Memory/ Conditioning
1.00–0.90	6	3	2	1	3	2	1		1
0.89–0.80	7					1			
0.79–0.70	7	2	1	1	3				
0.69–0.60	4	1	1		1			1	
0.59–0.50	3								
0.49–0.40	3				1	1			
0.39–0.30									
0.29–0.20	1								

**Table 5 ijms-24-03040-t005:** DEGs encoding transcription factors that make the largest contribution (r > |0.90|) to C_AD differences.

Transcription Factor Genes	Gene Expression, FPKM		log_2_ (Fold Change) AD/C	q_Value	Correlation between Gene Expression and PLS-DA Axis 1	Gene Name
	Control	AD				
*Nr1d1*	64.02	78.53	0.29	0.002	−0.968	Nuclear receptor subfamily 1, group D, member 1
*Fmr1*	19.09	15.37	−0.31	0.002	0.956	Fragile X messenger ribonucleoprotein 1
*Creb1*	7.61	5.75	−0.41	0.002	0.942	cAMP responsive element binding protein 1

**Table 6 ijms-24-03040-t006:** Behavioral-trait–associated candidate genes that contribute the most to C_AD differences.

Correlation between DEGs’ Expression and PLS-DA Axis 1 |r|	Abnormal Aggression- Related Behavior	Increased Aggression	Abnormal Anxiety-Related Response	Abnormal Fear-Related Response	Abnormal Response to Social Novelty	Learning or Memory	Abnormal Learning/ Memory/ Conditioning
1.00–0.95			*Nr1d1 **				
	*Fmr1 **	*Fmr1 **	*Fmr1 **	*Fmr1 **	*Fmr1 **		*Fmr1 **
0.94–0.90	*Pten*		*Pten*	*Pten*			

* Transcription factor genes.

**Table 7 ijms-24-03040-t007:** Top 10 C_AD DEGs whose expression correlates with the expression of key hippocampal DEGs associated with neurogenesis and behavioral traits of experimental male mice.

Key C_AD DEGs Related to Neurogenesis and Behavioral Traits					
*Nr1d1*		*Fmr1*		*Pten*	
Gene Symbol	r	Gene Symbol	r	Gene Symbol	r
*Nufip2*	−0.973	*Tmem229a*	0.975	*Eif5a2*	0.975
*Sacm1l*	−0.970	*Lztfl1*	0.962	*Pcm1*	0.973
*Xpo1*	−0.969	*Pcm1*	0.961	*Dzip3*	0.973
*Pcmtd2*	−0.968	*Abcb7*	0.960	*Pum2*	0.972
*Hnrnpc*	−0.968	*Spred1 **	0.958	*Lnp*	0.972
*9330159F19Rik*	−0.965	*Pten *^#§^*	0.955	*Vcpip1*	0.971
*Tmem33*	−0.962	*Azin1*	0.954	*Rlim*	0.971
*Pcnp*	−0.961	*Rlim*	0.954	*Elavl2 ^©^*	0.968
*Picalm*	−0.960	*Far1*	0.954	*Taok1*	0.968
*Rexo1*	0.959	*Dzip3*	0.953	*Fndc3a*	0.968

* Abnormal anxiety-related response; ^#^ abnormal aggression-related behavior; ^§^ abnormal fear-related response; ^©^ transcription factor gene; r: Pearson correlation coefficient.

## Data Availability

All relevant data are available in Appendix A and from the authors.

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
