# Peer review of "Effects of Positive Fighting Experience and Its Subsequent Deprivation on the Expression Profile of Mouse Hippocampal Genes Associated with Neurogenesis"

_ijms, 2023, doi:10.3390/ijms24033040_

Round 1
Reviewer 1 Report
The manuscript entitled “Influence of Positive Fighting Experience and Its Deprivation on the Expression Profile of Mouse Hippocampal Genes Associated with Neurogenesis” addresses the identification of the expression profile of neurogenesis-linked hippocampal genes in mice with a chronic experience of wins in aggressive confrontations and during the subsequent 2-week deprivation from agonistic interactions. Herein, expression levels of neurogenesis genes were characterized in the experimental groups. Some of these genes were linked to behavioral traits in experimental males, including abnormal aggression-related behavior and abnormal anxiety-related response. Notably, the two genes encoding transcription factors (Nr1d1, Fmr1) represent a major contribution. The current findings revealed that chronic experience of wins in agonistic interactions triggered alterations in the transcription of several genes in the hippocampus of adult male mice. Interestingly, the transcriptome changes are only partially restored after a two-week period of deprivation from fights.
The current findings are interesting, and the manuscript is clearly written.
Comments:
1) Regarding RNA-Seq data collection, have the authors considered that mRNA gene expression assays using RT-PCR may not be adequate for quantifying the target signals since the mRNA expression may not necessarily reflect the corresponding protein levels due to the post-translational modifications?
2) In Section 4.3., mRNA preparation is missing the technical repeat information (whether each sample was repeated during the assay). Moreover, did the authors check the RNA quality with A260/280, and perform an RT negative control to ensure no DNA contamination in the RNA extraction? Please, add these data in section 4.3. in the material and methods.
3) In Section 4.3., the author should mention the amount of RNA used to synthesize cDNA and the amount of cDNA used for qRT-PCR part. Please, add these data in section 4.3. in the material and methods.
4) The authors are advised to add the cat. no. for the used kits and reagents.
5) To make all figure legends stand-alone and make it clearer for readers, the authors are advised to add the full name of the used abbreviations at the end of each legend. In the figure legend, the authors are also advised to elaborate on the description of how each of the 3 experimental groups was treated (control, A20, and AD).
6) In line 600 (Author contributions), the authors are advised to add only the initial letters for the name of each author rather than the full name, as stipulated by the journal.
Author Response
The manuscript entitled “Influence of Positive Fighting Experience and Its Deprivation on the Expression Profile of Mouse Hippocampal Genes Associated with Neurogenesis” addresses the identification of the expression profile of neurogenesis-linked hippocampal genes in mice with a chronic experience of wins in aggressive confrontations and during the subsequent 2-week deprivation from agonistic interactions. Herein, expression levels of neurogenesis genes were characterized in the experimental groups. Some of these genes were linked to behavioral traits in experimental males, including abnormal aggression-related behavior and abnormal anxiety-related response. Notably, the two genes encoding transcription factors (Nr1d1, Fmr1) represent a major contribution. The current findings revealed that chronic experience of wins in agonistic interactions triggered alterations in the transcription of several genes in the hippocampus of adult male mice. Interestingly, the transcriptome changes are only partially restored after a two-week period of deprivation from fights.
The current findings are interesting, and the manuscript is clearly written.
Comments:
1) Regarding RNA-Seq data collection, have the authors considered that mRNA gene expression assays using RT-PCR may not be adequate for quantifying the target signals since the mRNA expression may not necessarily reflect the corresponding protein levels due to the post-translational modifications?
Answer: You are right, the mRNA expression may not necessarily reflect the corresponding protein levels. Nonetheless, as shown in many studies, transcriptomic analysis allows investigators to determine the main molecular events in cells and to understand which specific genes and metabolic pathways are most significantly involved in the adaptation of the body to effects of endogenous or exogenous factors. In connection with this comment, a corresponding discussion has been made in the text of the manuscript (Lines 505-513).
2) In Section 4.3., mRNA preparation is missing the technical repeat information (whether each sample was repeated during the assay). Moreover, did the authors check the RNA quality with A260/280, and perform an RT negative control to ensure no DNA contamination in the RNA extraction? Please, add these data in section 4.3. in the material and methods.
Answer: All the required information is added to the section 4.3. of the manuscript.
3) In Section 4.3., the author should mention the amount of RNA used to synthesize cDNA and the amount of cDNA used for qRT-PCR part. Please, add these data in section 4.3. in the material and methods.
Answer: Done
4) The authors are advised to add the cat. no. for the used kits and reagents.
Answer: Done
5) To make all figure legends stand-alone and make it clearer for readers, the authors are advised to add the full name of the used abbreviations at the end of each legend. In the figure legend, the authors are also advised to elaborate on the description of how each of the 3 experimental groups was treated (control, A20, and AD).
Answer: Done
6) In line 600 (Author contributions), the authors are advised to add only the initial letters for the name of each author rather than the full name, as stipulated by the journal.
Answer: Done
The authors are grateful to the reviewer for constructive comments, which allowed us to strengthen the manuscript.

Reviewer 2 Report
Redina et al., aimed to investigate the phenotypic characteristics of male mice with chronic positive fighting experience in daily agonistic interactions as well as with a subsequent two-week deprivation period as compared to a control group by identifying differentially expressed genes associated with changes in hippocampal neurogenesis and thought to be involved in the manifestation of the exhibited behavioral features. The authors previously demonstrated that neurogenesis increases in the hippocampus of chronically victorious male mice with these changes remaining mostly intact after a period of deprivation, and therefore they worked to identify the genetic framework driving this phenomenon at the transcriptional level. The impact of an experience on neurogenesis and characterization of the related transcriptional changes has is of importance, however, there are some major concerns that impact the interpretation of the current work, as well as further analyses that would enhance the interpretation.
Major concerns:
The authors have not described the hippocampal dissections in any capacity and they need to do so. The description should include the subregions of the hippocampus that were included in the dissection and the landmarks used to delineate the regions included in the dissection. Was the dissection limited to dentate gyrus? Were other subregions included? Were dissections done from slices? Whole brain? Dorsal hippocampus only? Dorsal and ventral hippocampus? Were samples taken from both hemispheres? If only one hemisphere, was the same hemisphere used for all animals (and which hemisphere)?
Because the samples collected likely included a heterogenous sampling of cells, including neurons, interneurons, oligodendrocytes, etc., it would be useful for the authors to consider performing some type of cell deconvolution to look for enrichment of markers of various cell types, other than just the markers of enhanced neurogenesis. This should be considered even if the authors believe they have restricted their samples to the dentate gyrus. Such an analysis could shed a lot of light on the mechanisms involved in neurogenesis-related changes, in particular in how these new neurons are incorporated into the network.
There is also indication of the RNA quality. Was RNA quality checked? If so, how? Were RNA integrity numbers assessed? Was there a cut off for a sample to be included?
It is not clear what the control mice experience. What happens during the 5 days for these control mice? Additionally, the control isn’t a convincing control. The control mice should have had the same number of days of manipulation, albeit without a partner mouse, to account for the experimental manipulations, or have been a home cage control with no manipulations. The authors should briefly indicate why they chose this design as control mice.
The authors argue that the DEGs indicate the mice experience heighted stress following aggression. Was there ever a direct measure of stress for this experimental group, such as measurement of cortisol in the blood?
The authors posit that there was enrichment of DEGs in behavior-related terms that most appropriately characterize the behavior of the chronically winning mice: abnormal aggression-related behavior, abnormal anxiety-related response, abnormal fear-related response, and abnormal response to social novelty.
1) Do the DEGs indicate an increased directionality for these terms in the chronic fighting mice? It is possible these pathways would be predicted to be decreased based on the combination of DEGs being increased or decreased. Was directionality of differential expression taken into account for these clusters of genes?
2) References for these behaviors being present in the chronic fighting mice in this section would be useful.
The authors also suggest that the DEGs coexpressed with these genes may also be involved in the manifestation of the traits they are responsible for, even though they have yet to establish an association with the behavioral traits being examined. How can they surmise this if they haven’t established the associations? Is it indicated in any of the supplementary materials? If so, where?
The authors should also comment on the timing of neurogenesis, which are likely lagging approximately two weeks behind the experiences which are driving the increases they observe.
Minor concerns:
It is unclear to me why the authors think the aggression-conditioned mice have abnormal learning and memory. Has this been examined directly in a previous study?
Looking at the principal component analysis, there is a lot of variability in PC2, which makes me wonder if the level of aggressive behavior (number of bites, tail rattles, etc.) could be correlated to gene expression, and if the degree of aggression was previously correlated to the amount of neurogenesis (although I recognize that there may be a ceiling effect in the aggression level).
When talking about the differential gene expression of the deprivation group, the authors say that: “Therefore, it can be assumed that some of them require a longer period of time to restore the control level of transcription.” It would be more thorough to indicate that some of these DEGs may never return to baseline levels.
The use of the word “positive” to describe the fighting experience implies that being victorious is somehow good for the mice, but the authors indicate that the experience is stressful and leads to increased anxiety and abnormal learning and memory.
According to the authors, under these experimental conditions, significant changes in the level of transcription of ribosomal genes in different parts of the brain were detected. It would be useful if the authors touch more upon the ramifications of the activation of hippocampal ribosomal genes in particular, and how their experiments might be looked at from a more translational level.
There are a number of grammatical issues (see below for examples, but there may be more):
- In the first sentence of the abstract it says “The hippocampus is known as brain region implicated in visuospatial processes…” Grammatically, this should be written as “as the brain region.”
- The fourth paragraph of the introduction begins with “We have also studied some parameters of neurogenesis and neuronal activity in mice with positive fighting experience in a chronic social conflicts.” This should either say “in a chronic social conflict” or “in chronic social conflicts.”
- Section 1.2 of the results section ends with saying “8 of which reduce the level of transcription under aggression experience accompanying by wins.” This should say “accompanied by wins” instead.
- Section 1.3 of the results section is titled, and begins with, “To identify DEGs associated with neurogenesis and making the greatest contribution to intergroup C_A20 differences…” Grammatically, this would be better written as “To identify DEGs associated with neurogenesis and that make the greatest contribution…”
- In the sixth paragraph of the discussion, it says “Based on the clinical observations it was shown that the patients with mood disorders often demonstrate the abnormalities in the circadian rhythms which was associated…” This should say “which were associated…”
- In the sixth paragraph of the discussion, it says “Based on that observations…” This should say “Based on those observations…”
- In the eighth paragraph of the discussion, it says “Earlier it was reported that Nr1d1 induced in mouse striatum by chronic drug administration has most substantial impact on the gene expression profile.” This should say “Earlier it was reported that Nr1d1 induced in the mouse striatum by chronic drug administration has the most substantial impact on the gene expression profile.”
- The sentence immediately preceding reference 51 needs to be edited for clarity. The portion of the sentence that says “but its activation in brain areas that are usually not associated with neurogenesis processes does not negate the involvement of Nr1d1 in them” is particularly confusing and it is not readily apparent what exactly “its activation” and “in them” are referring to.
- The sentence immediately preceding reference 52 says “FMRP regulates gene expression and the translation of multiple mRNAs playing important role in the development…” This should either say “multiple mRNAs playing important roles” or “multiple mRNAs playing an important role.”
Author Response
Redina et al., aimed to investigate the phenotypic characteristics of male mice with chronic positive fighting experience in daily agonistic interactions as well as with a subsequent two-week deprivation period as compared to a control group by identifying differentially expressed genes associated with changes in hippocampal neurogenesis and thought to be involved in the manifestation of the exhibited behavioral features. The authors previously demonstrated that neurogenesis increases in the hippocampus of chronically victorious male mice with these changes remaining mostly intact after a period of deprivation, and therefore they worked to identify the genetic framework driving this phenomenon at the transcriptional level. The impact of an experience on neurogenesis and characterization of the related transcriptional changes has is of importance, however, there are some major concerns that impact the interpretation of the current work, as well as further analyses that would enhance the interpretation.
Major concerns:
The authors have not described the hippocampal dissections in any capacity and they need to do so. The description should include the subregions of the hippocampus that were included in the dissection and the landmarks used to delineate the regions included in the dissection. Was the dissection limited to dentate gyrus? Were other subregions included? Were dissections done from slices? Whole brain? Dorsal hippocampus only? Dorsal and ventral hippocampus? Were samples taken from both hemispheres? If only one hemisphere, was the same hemisphere used for all animals (and which hemisphere)?
Answer: All hippocampal subregions from both hemispheres were collected. We added this information to the text of manuscript (Lines 573-574).
- Because the samples collected likely included a heterogenous sampling of cells, including neurons, interneurons, oligodendrocytes, etc., it would be useful for the authors to consider performing some type of cell deconvolution to look for enrichment of markers of various cell types, other than just the markers of enhanced neurogenesis. This should be considered even if the authors believe they have restricted their samples to the dentate gyrus. Such an analysis could shed a lot of light on the mechanisms involved in neurogenesis-related changes, in particular in how these new neurons are incorporated into the network.
Answer: The reviewer's suggestion is certainly very correct. However, we have deliberately devoted this article only to the known genes involved in neurogenesis, since this topic in the study of the hippocampus under conditions of social conflict (stress) is special due to the fact that the hippocampus is considered one of the most important brain regions in which adult neurogenesis is observed. These limitations allowed us to examine in detail the genes of neurogenesis and devote the entire article to their discussion. Most of the DEGs found in our study are not considered in this manuscript, since, according to the analysis of the changes in the entire transcriptome profile in the hippocampus of A20 and AD mice compared with the control, a separate manuscript is currently being prepared, where all the issues that the reviewer writes about will be considered in detail.
- There is also indication of the RNA quality. Was RNA quality checked? If so, how? Were RNA integrity numbers assessed? Was there a cut off for a sample to be included?
Answer: the details of RNA processing and quality are added to the text (section 4.3. in the material and methods; lines 580-588).
- It is not clear what the control mice experience. What happens during the 5 days for these control mice? Additionally, the control isn’t a convincing control. The control mice should have had the same number of days of manipulation, albeit without a partner mouse, to account for the experimental manipulations, or have been a home cage control with no manipulations. The authors should briefly indicate why they chose this design as control mice.
Answer: The logic of applying our control was as follows: When mice are housed in grouped cages the dominant-subordinate relationships develop between them – one of them is dominant and other mice are subordinates that do not demonstrate aggressive behavior in a conflict situation. Keeping males one per cage for 5 days allows us to obtain adequate control, since every mouse became dominant and all of them can demonstrate aggressive behavior in a conflict situation. These mice have no consecutive experience of aggression and serve as control.
The problem of the control in our model has been discussed many times. Please, see papers for details:
Kudryavtseva N.N. (1991) The sensory contact model for the study of aggressive and submissive behaviors in male mice. Aggress. Behav. 17(5), 285-291.
Kudriavtseva N.N. (1999) Agonistic behavior: a model, experimental studies, and perspectives. Neurosci. Behav. Physiol. 2000, 30(3):293-305.
Kudryavtseva N.N. Sensory contact model: Protocol, control, applications, Horizons in Neuroscience Research. NOVA Science Publishers Inc., New York, 2011, Editors: Andres Costa and Eugenio Villalba, V.3, Ch. 4, pp. 81-100 https://link.springer.com/content/pdf/10.1038/npre.2009.3299.1.pdf
The following text was added to the section 4.2. in the material and methods:
Control males were housed one per cage for 5 days which allows them to feel as dominant and potentially able to demonstrate aggressive behavior in a conflict situation (Lines 535-537)
- The authors argue that the DEGs indicate the mice experience heighted stress following aggression. Was there ever a direct measure of stress for this experimental group, such as measurement of cortisol in the blood?
Answer: The manuscript shows that many genes associated with both neurogenesis and stress response retain altered transcription levels in the mouse hippocampus after 14 days of fight deprivation (Figure 5).
A sensory contact model has been used for many years in various laboratories for studying the impact of psychosocial stress in mice. Using the keywords "sensory contact model" and “stress” opens more than 300 articles in the bibliographic database https://scholar.google.com. This demonstrates that not only the authors of this study, but also researchers from other scientific teams using this model, believe that animals are stressed.
- The authors posit that there was enrichment of DEGs in behavior-related terms that most appropriately characterize the behavior of the chronically winning mice: abnormal aggression-related behavior, abnormal anxiety-related response, abnormal fear-related response, and abnormal response to social novelty.
- Do the DEGs indicate an increased directionality for these terms in the chronic fighting mice? It is possible these pathways would be predicted to be decreased based on the combination of DEGs being increased or decreased. Was directionality of differential expression taken into account for these clusters of genes?
- References for these behaviors being present in the chronic fighting mice in this section would be useful.
Answer: Since we have a phenotypic assessment of experimental animals that are characterized by increased aggression, increased anxiety, and some signs of cognitive dysfunction, the terms associated with behavior that we have chosen, from our point of view, best describe the psychosocial state of the experimental mice in our study. These terms were used to carry out a functional annotation of the found DEGs.
The detailed description of winners’s behavior was presented in reviews which have been cited in the Introduction section of the manuscript:
Kudryavtseva, N.N.; Smagin, D.A.; Kovalenko, I.L.; Vishnivetskaya, G.B. Repeated positive fighting experience in male inbred mice. Nat Protoc 2014, 9, 2705-2717, doi:10.1038/nprot.2014.156. Kudryavtseva, N.N. Positive fighting experience, addiction-like state, and relapse: Retrospective analysis of experimental studies. Aggress. Viol. Behav. 2020, 52, 101403, doi:doi.org/10.1016/j.avb.2020.101403.
The pattern of changes in the transcription level of DEG clusters in the hippocampus obtained in our study corresponds to a state of increased aggression, increased anxiety, and some deviations in the behavior associated with learning and memory in experimental mice [33, 35]. Having identified the most likely genes that make the greatest contribution to intergroup differences, we tried to analyze their contribution to the phenotype, taking into account the direction of change in their transcription in the hippocampus. However, we understand that the phenotype of experimental mice, of course, is formed as a set of changes in transcription profiles in different brain structures. At the same time, it should be kept in mind that in different regions of the brain, changes in the level of transcription of the same DEG can change in different directions, which was described in several of our previously published articles. We have added these considerations to the discussion text.(lines 486-495).
- The authors also suggest that the DEGs coexpressed with these genes may also be involved in the manifestation of the traits they are responsible for, even though they have yet to establish an association with the behavioral traits being examined. How can they surmise this if they haven’t established the associations? Is it indicated in any of the supplementary materials? If so, where?
Answer: This point has been discussed in the manuscript (lines 475-480). The authors included this part of the analysis in the text of the manuscript, as they believe that this hypothesis, based on the correlation analysis, may be valid, and the information obtained may be useful for other groups of researchers studying the molecular genetic basis of the processes associated with the functions of the hippocampus.
- The authors should also comment on the timing of neurogenesis, which are likely lagging approximately two weeks behind the experiences which are driving the increases they observe
Answer: We made clarifications commenting on the timing of neurogenesis, based on the data we published earlier (lines 347-348) and expanded the discussion (lines 501-504).
Minor concerns:
- It is unclear to me why the authors think the aggression-conditioned mice have abnormal learning and memory. Has this been examined directly in a previous study?
Answer: We do have behavioral observations demonstrating cognitive dysfunction in winners. For example, the winners have a broken social recognition: they equally attack females, young male, male demonstrating submissive behavior. Intact males never do this. Also, the winners have impaired exploratory activity compared to intact mice [Lipina T.V., Kudryavtseva N.N. (2008) The study of explorative behavior in CBA/Lac male mice under influence of positive and negative social experience. Zh. Vyssh. Nerv. Deiat. im I.P. Pavlova 58(2), 194-201 (Ru)].
The category of abnormal learning and memory was taken into consideration because the functions of the hippocampus are primarily associated with the processes of learning and memory, and as we indicated above, the winners develop stable deviations in behavior as a result of chronic confrontations.
- Looking at the principal component analysis, there is a lot of variability in PC2, which makes me wonder if the level of aggressive behavior (number of bites, tail rattles, etc.) could be correlated to gene expression, and if the degree of aggression was previously correlated to the amount of neurogenesis (although I recognize that there may be a ceiling effect in the aggression level).
Answer: We cannot correlate changes in expression of neurogenesis genes in the hippocampus with number of bites, attacks tail rattling and so on.
- When talking about the differential gene expression of the deprivation group, the authors say that: “Therefore, it can be assumed that some of them require a longer period of time to restore the control level of transcription.” It would be more thorough to indicate that some of these DEGs may never return to baseline levels.
Answer: Thanks. This is a fair remark. The phrase that some of these DEGs may never return to baseline levels has been added to the text of the manuscript (see line 485).
- The use of the word “positive” to describe the fighting experience implies that being victorious is somehow good for the mice, but the authors indicate that the experience is stressful and leads to increased anxiety and abnormal learning and memory.
Answer: In our experimental context, the word “positive” has a meaning - accompanied by victory, positive reinforcement, not by defeat. Unfortunately, any positive reinforcement can lead to a negative effect (drug abuse, food abuse, sex abuse, gambling addiction, so on).
- According to the authors, under these experimental conditions, significant changes in the level of transcription of ribosomal genes in different parts of the brain were detected. It would be useful if the authors touch more upon the ramifications of the activation of hippocampal ribosomal genes in particular, and how their experiments might be looked at from a more translational level.
Answer: In connection with this comment, a corresponding discussion has been made in the text of the manuscript (Lines 505-513).
- There are a number of grammatical issues (see below for examples, but there may be more)
Answer: The grammatical errors indicated by the reviewer have been corrected and the text as a whole was corrected by shevchuk-editing.com. The language certificate is enclosed.
The authors are grateful to the reviewer for constructive comments, which allowed us to substantially strengthen the manuscript.
